# Temperature and Stagnation Effects on Ozone Sensitivity to $NO_x$ and VOC: An Adjoint Modeling Study in Central California

Yuhan Wang[1], Lucas Bastien[2,3], Yuan Wang[1], Ling Jin[3], and Robert Harley[4]

[1]Department of Earth System Science, Stanford University, Stanford, CA 94305, USA
[2]Univ. Grenoble Alpes, CNRS, INRAE, IRD, Grenoble INP*, IGE, 38000 Grenoble, France
[3]Energy Technologies Area, Lawrence Berkeley National Laboratory, Berkeley, CA 94720, USA
[4]Department of Civil and Environmental Engineering, University of California, Berkeley, CA 94720, USA

*Correspondence to*: Ling Jin (ljin@lbl.gov) and Robert Harley (harley@ce.berkeley.edu)

**Abstract.** Extreme weather events like heatwaves and stagnation are increasing with climate change. While their effects on ozone levels have been extensively studied, how extreme weather alters $O_3$-$NO_x$-VOC sensitivity and optimal mitigation strategies is less explored. Here, we apply the CMAQ adjoint model over central California to quantify ozone sensitivity to spatiotemporally resolved precursor emissions under three meteorological scenarios (baseline, high-T, and stagnation) and three emission years (2000, 2012, and 2022). Results show that meteorology-induced changes in sensitivity are comparable in magnitude to those from decadal emission reductions. Higher temperature (+5 °C) amplifies ozone sensitivity to both $NO_x$ and VOC, with the largest relative increase in biogenic VOC sources. High-T conditions shift ozone chemistry toward $NO_x$ limitation under a VOC-limited emission scenario, but increase the relative importance of VOC control for a $NO_x$-limited scenario. Stagnation consistently pushes ozone chemistry toward VOC limitation across emission scenarios, increasing VOC sensitivity by a factor of ~3-4. Stagnation also spatially shifts influential source areas, especially for $NO_x$, and temporally amplifies prior-day emission impacts due to enhanced pollutant carryover. As the study domain transitions to a $NO_x$-limited regime over time, we identify a growing subset of "climate-resilient" source targets that remain impactful across meteorological scenarios, along with spatial convergence in optimal locations for $NO_x$ and VOC emission control. These findings underscore both the need and feasibility to consider meteorological extremes in the design of ozone mitigation strategies for a warming climate.

## 1 Introduction

Heatwaves and atmospheric stagnation events are increasing in frequency, intensity, and duration due to climate change (Meehl and Tebaldi, 2004; Horton et al., 2014; Hou and Wu, 2016). Between 1951-1980 and 1981-2010, the occurrence of heatwaves and stagnation events increased by 25.8±3.3% and 4.5±0.8%, respectively, for non-polar continental regions (Hou and Wu, 2016). Meteorological conditions modulate air pollution by changing weather-dependent emissions (e.g., from biogenic sources), regulating the transport and removal of emitted pollutants, and altering temperature- and light-sensitive chemical reaction rates. Although extremely high temperature may suppress ozone ($O_3$) formation (Steiner et al., 2010), most evidence robustly link heatwaves and stagnation events to elevated ozone pollution (Gao et al., 2013; Hou and Wu, 2016; Shen et al., 2016; Schnell and Prather, 2017; Zhang et al., 2018). Given ozone's adverse impacts on human health and ecosystems (Jerrett et al., 2009; Krupa et al., 2001), effective ozone mitigation under these pollution-favoring conditions is critical.

Ground-level ozone forms through photochemical reactions involving its precursors: nitrogen oxides ($NO_x$) and volatile organic compounds (VOC). This $O_3$-$NO_x$-VOC chemistry is complex and highly nonlinear (Pusede et al., 2015; Kleinman, 2005). In environments with high VOC/$NO_x$ ratios (*$NO_x$-limited regime*), ozone formation is primarily controlled by $NO_x$ availability and relatively insensitive to VOC changes. Conversely, when the VOC/$NO_x$ ratio is low (*VOC-limited regime*), controlling VOC is

more effective; $NO_x$ reductions may increase hydrogen oxide ($HO_x$) radicals and decrease ozone titration in this regime, thereby exacerbating ozone pollution. A *transition regime* exists between $NO_x$- and VOC-limited conditions. Across these regimes, ozone responses to precursor reductions can vary greatly in magnitude and sign, depending on where, when, and which precursor emissions are reduced. Knowledge about $O_3$-$NO_x$-VOC sensitivities is therefore key to identifying and controlling emissions from the most influential sources. Although meteorological controls on ozone concentrations are well studied, far less work has examined how meteorology modulates $O_3$-$NO_x$-VOC sensitivities and thus the effectiveness of emission control programs.

Some studies have explored how warming shifts ozone sensitivities and their conclusions vary. Baertsch-Ritter et al. (2004) perturbed temperature by up to +10 °C in a 3D air quality model for northern Italy, finding that VOC-limited areas expand with higher temperature, primarily because it promotes peroxyacetyl nitrate (PAN) decomposition and increases $NO_x$ availability. Similarly, Jin et al. (2010) found increased sensitivity to VOC and decreased sensitivity to $NO_x$ under a +2 °C temperature perturbation for the San Francisco Bay area. They attributed these sensitivity changes to elevated radical generation, which amplifies VOC oxidation and accelerates $NO_x$ removal. In contrast, some observational studies have inferred that ozone chemistry becomes more $NO_x$-limited at high temperatures. Yang et al. (2021) and Huang et al. (2025) analyzed measurements in Shanghai and the Pearl River Delta in China, and diagnosed an ozone regime switch from primarily VOC-limited to transitional or $NO_x$-limited when exceeding a region-specific temperature threshold. Studies in California's San Joaquin Valley and Los Angeles air basins found strong temperature dependencies for some VOC sources, such as evaporative and biogenic processes (Pusede and Cohen, 2012; Pusede et al., 2014; Nussbaumer and Cohen, 2020; Pfannerstill et al., 2024). They pointed out that VOC reactivity rises with temperature, whereas $NO_x$ reactivity is relatively temperature-invariant. On hotter days, observed increases in weekend $O_3$ relative to weekdays tend to be attenuated (this suggests a shift to a more $NO_x$-limited regime), though some sites and years show opposite trends (increasingly VOC-limited conditions). These contrasting findings likely arise from differences in baseline chemical regimes, VOC source profiles, and the temperature ranges examined.

Even fewer studies have quantified the effects of stagnation events on ozone sensitivity. Prior modeling evidence generally suggests that ozone formation regimes tend to become more VOC-limited when wind speeds, boundary layer heights, and humidity are low – all are characteristic features of stagnation events (Biswas and Rao, 2001; Baertsch-Ritter et al., 2004; Jin et al., 2008, 2010, 2013). Biswas and Rao (2001) argued that in well-ventilated environments, $NO_x$ is removed more rapidly than VOC as air masses age photochemically, leading to a shift toward $NO_x$-limited conditions. In contrast, weak dilution (slow wind speeds and shallow boundary layer heights) limit $NO_x$ removal and maintain more VOC-limited conditions. Baertsch-Ritter et al. (2004) also report a slight expansion of VOC-limited areas under low-humidity conditions, though the water vapor influence is expected to be weak overall. Jin et al. (2013) simulated a stagnation event in California's San Joaquin Valley, incorporating changes in all meteorological variables and biogenic emissions. Using perturbation-based sensitivity analysis, they showed that stagnation enhances the relative importance of VOC control and reduces the influence of upwind sources. As stagnation affects both transport patterns and chemical regimes, the most relevant sources for effective ozone mitigation are also expected to shift, but systematic investigations of such impacts are lacking.

Most of the prior studies use either observations or model perturbations to derive ozone sensitivity and formation regime. Observation-based diagnostics include ground-based indicators like ozone production efficiency ($\Delta O_3/\Delta NO_x$) and total reactive nitrogen (Liu et al., 1987; Milford et al., 1994), satellite-retrieved proxies such as $HCHO/NO_2$ ratios (Jin et al., 2017, 2020), and the sign of $O_3$ weekday-weekend difference as a natural experiment (Marr and Harley, 2002; De Foy et al., 2020). Additionally, metrics such as relative incremental reactivity and ozone isopleth diagrams can be derived using observation-constrained box models (Dodge, 1977; Cardelino and Chameides, 1995; Guo et al., 2023). As these methods rely on localized observations, they

tend to neglect the effects of regional pollutant transport and mesoscale meteorology. The use of 3D chemical transport models addresses these limitations. In these models, ozone sensitivity can be evaluated through either brute-force emission perturbations or more sophisticated sensitivity analysis techniques, such as *the adjoint method* used in this study. As an advanced receptor-oriented sensitivity analysis technique, the adjoint method can accurately compute the sensitivities of any user-defined air quality metric (*receptor*) with respect to emission changes at each grid cell, time step, and for each precursor species. The adjoint-based $O_3$-$NO_x$-VOC sensitivities are particularly policy-relevant, as they can directly pinpoint the most relevant sources for ozone receptors at urban, regional, and national scales (Pappin and Hakami, 2013; Wang et al., 2022, 2023; Hu et al., 2024).

To our knowledge, no prior studies have systematically investigated meteorological impacts on adjoint-based $O_3$-$NO_x$-VOC sensitivities. This is a second-order sensitivity analysis: how ozone sensitivity, rather than ozone itself, responds to meteorological changes. Existing adjoint literature has laid important foundations at the first-order level. Hakami et al. (2007) evaluated ozone exposure sensitivity to local temperatures through chemical kinetics. Zhao et al. (2013) advanced this work to estimate ozone climate penalties in Canada and the U.S. using adjoint-derived temperature sensitivities, considering both direct chemical kinetic impacts and indirect effects through biogenic emissions and water vapor. They found high-$NO_x$ polluted regions were particularly responsive to temperature. Park et al. (2018) assessed $O_3$-$NO_x$-VOC sensitivities in Daegu, Korea, attributing 62% of total contributions to meteorology-driven $O_3$ transport and 38% to chemical reactions. Ashok and Barrett (2016) constructed ozone exposure isopleths in the U.S. using adjoint sensitivities, with discussion of seasonal variability bearing relevance to second-order impacts. While these studies highlight the potential for meteorology to affect ozone regimes and emission control strategies, our work seeks to directly and systematically isolate this effect across representative meteorological regimes.

In this study, we assess how changes in meteorology alter $O_3$-$NO_x$-VOC sensitivity and optimal control strategies. We apply the adjoint method with a 3D chemical transport model to the San Joaquin Valley in central California, to quantify sensitivities of regional ozone formation to precursor emission changes across chemical species, space, and time. We analyze three distinct meteorological scenarios, each representing a physically coherent and historically observed episode: (1) typical summer conditions (*baseline*), (2) a high-temperature case representing a heatwave (*high-T*), and (3) an episode with atmospherically stagnant conditions (*stagnation*). We evaluate not only changes in ozone responses, but also how meteorology shifts the type, location, and timing of the most influential precursor emissions.

Three emission inventory timeframes (2000, 2012 and 2022) are used in this study to capture a variety of chemical conditions. The year 2000 represents a more VOC-limited environment, whereas year 2022 reflects cleaner, $NO_x$-limited conditions following major $NO_x$ emission reductions (Pusede and Cohen, 2012; De Foy et al., 2020; Wang et al., 2023). This study design allows us to investigate how meteorological effects vary across different chemical regimes, and how such influences may evolve on decadal timescales as emissions decline. Through this multi-scenario, multi-year framework, our findings offer insights into how ozone mitigation strategies should adapt in a warming climate.

## 2 Methodology

### 2.1 Modeling Domain

The San Joaquin Valley (SJV) and adjacent air basins in central California (34.5-39°N, 118.5-123°W) are shown in Figure 1. The SJV is among the worst polluted regions in the United States, and is designated as an extreme ozone nonattainment area. The national standard for 8-hour ozone concentrations is 70 ppb, whereas SJV's design value has exceeded 85 ppb as of 2024, even

after excluding wildfire impacts (San Joaquin Valley Air Pollution Control District, 2025). The poor air quality poses health concerns for over four million SJV residents.

Land use in the SJV is characterized by widely extended agricultural activity, with major cities distributed along the main traffic corridor located on the east side of the valley (Highway 99, see Figure 1). The valley is bordered by the Sierra Neveda Mountains to the east and Coastal Mountain range to the west (Figure S1). On typical summer days, emissions of ozone precursors from upwind source regions including the San Francisco Bay area (SF Bay) and the Sacramento Valley (SV) are transported into the SJV. Within the valley, the prevailing wind direction is generally from the northwest to southeast.

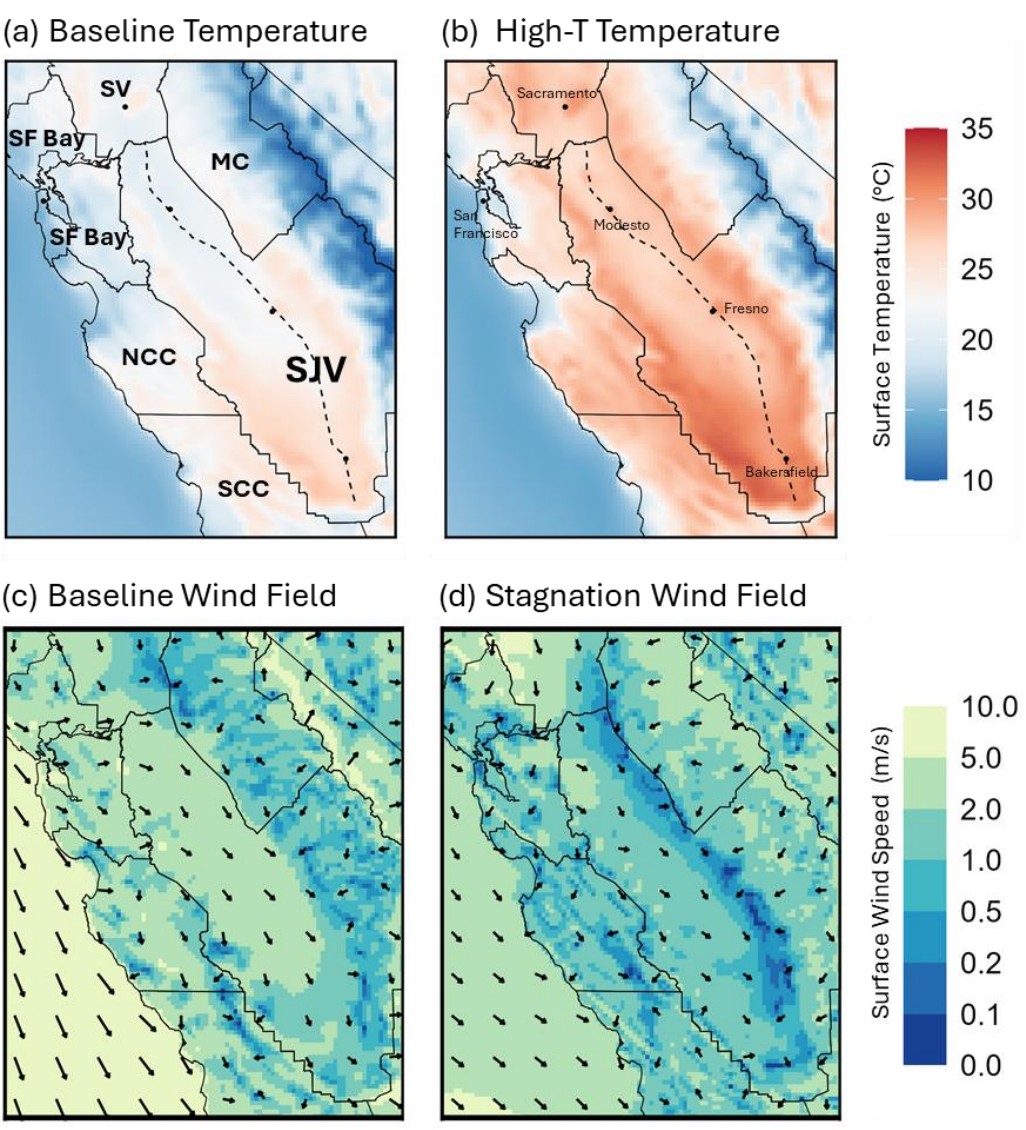

**Figure 1.** Daily average surface temperature (a-b) and wind fields (c-d). Air basins labeled in panel (a) include San Joaquin Valley (SJV), San Francisco Bay area (SF Bay), Sacramento Valley (SV), Mountain Counties (MC), North Central Coast (NCC) and South Central Coast (SCC). Major cities are shown in panel (b), with the dashed line representing Highway 99.

## 2.2 Meteorological Scenarios

Three meteorological scenarios are considered in this study: baseline, high-T, and stagnation. Each scenario encompasses a weeklong ozone episode (Table 1). These episodes are selected based on Jin et al. (2011), who applied K-means clustering to hourly meteorology and ozone fields across central California for an entire summer. Their analysis identified six statistically distinct meteorological regimes, each associated with unique ozone spatial distributions and pollution levels. From these, we selected the three regimes that favor ozone formation and extracted typical meteorological conditions for weeklong timeframes within each regime. The three regimes differ primarily either in temperature (baseline vs. high-T), or in flow patterns (baseline vs. stagnation), with simultaneous changes in interconnected meteorological variables captured. We acknowledge that compound extremes (e.g., concurrent heat and stagnation) are also of growing importance (Gao et al., 2013; Zhang et al., 2018), but our scenario selection is designed to capture statistically distinct regimes rather than their overlap. Explicit evaluation of compound effects is beyond the scope of this study and is a recommended focus for future work. This cluster-based approach offers a few advantages. First, it preserves the physical coherence of meteorological fields, in contrast to artificial perturbations of individual variables. Second, these regimes are seen as recurring features across summers, so our scenario design has implications beyond one single year. Our findings can help anticipate how climate-induced changes in the frequency and intensity of such regimes will alter $O_3$-$NO_x$-VOC sensitivities and control effectiveness over time.

Gridded meteorological fields at 4 km resolution are simulated for the baseline, high-T, and stagnation scenarios using their corresponding episodes (Table 1) in the fifth-generation Penn State/NCAR Mesoscale Meteorological Model (MM5; Grell et al., 1994). These hourly fields – including wind, temperature, pressure, boundary layer dynamics, and radiation – are subsequently used as inputs for air quality modeling and sensitivity evaluation. Table 1 summarizes daily average surface temperatures and wind speeds, as well as daytime (10am–6pm) PBL heights under each scenario. Figure 1 contrasts surface temperature fields between baseline and high-T cases, and wind fields between baseline and stagnation. Diurnal evolutions of meteorological fields are shown in Supporting Figures S2–S4.

**Table 1**. Meteorological scenarios used in the study.

| Scenario | Episode | Daily T (°C)* | Daily Wind (m/s)* | 10am-6pm PBL (m)* |
|---|---|---|---|---|
| Baseline | 11 – 17 August, 2000 | 24 | 2.3 | 600 |
| High-T | 27 July – 2 August, 2000 | 29 | 1.8 | 622 |
| Stagnation | 14 – 20 September, 2000 | 26 | 1.1 | 467 |

\* Values shown are averaged across the SJV air basin (excluding mountain grid cells)

The *baseline* scenario features relatively ventilated conditions (daily mean wind speed ~2.3 m/s) and a moderate daily average temperature of ~24°C (Figure 1a). Strong westerly winds along with inflows from the north carry ozone precursors into the SJV and transport them southward along its axis (Figure 1c). The 10am-6pm planetary boundary layer (PBL) height is ~600 m, indicative of moderate vertical mixing conditions.

The *high-T* scenario is characterized by a +5°C temperature anomaly relative to baseline (Figure 1b). Its flow patterns and PBL heights resemble the baseline (Figures S2-S3), although wind speeds are ~20% slower. This meteorological regime typically corresponds to the presence of a persistent high-pressure ridge over the western U.S., which promotes subsidence, suppresses cloud cover, and results in consecutive days of elevated surface temperatures.

The *stagnation* scenario represents an atmospheric stagnation event. Its conditions satisfy commonly used stagnation criteria, including weak surface and mid-troposphere winds, and minimal precipitation (Wang and Angell, 1999). Compared to baseline, this scenario exhibits 53% and 22% reductions in average wind speeds and PBL heights, respectively (Table 1, Figure 1d). Although a modest (+2°C) temperature anomaly is part of this scenario (Figure S4), the primary driver of sensitivity changes in this case is attributed to stagnation rather than temperature effects. Jin et al. (2013) linked this meteorological regime with an anticyclone system in the Northeastern Pacific, which heats up air over the ocean and in coastal areas and weakens the pressure gradient that drives onshore flow. The resulting stagnation limits pollutant inflow from upwind areas and inhibits pollutant outflow at the southern end of the valley, thereby increasing pollutant residence times.

### 2.3 Data and Model Setup

The Community Multiscale Air Quality Model (CMAQ) version 4.5 and its adjoint are used in this study with the same configuration (Byun and Schere, 2006; Hakami et al., 2007; Bastien et al., 2015, 2019). The CMAQ model simulates atmospheric chemistry and transport processes, while CMAQ Adjoint computes the sensitivities of a user-defined metric (*receptor*) with respect to emission changes. Here, the receptor is defined to be population-weighted 8-hour average (10am-6pm) concentration of odd oxygen ($O_x = O_3 + NO_2$) within the SJV. We use $O_x$ rather than $O_3$ because $O_x$ is conserved under rapid NO-$O_3$ titration, providing a more stable measure of the photochemical oxidant burden for evaluating its sensitivity to precursors.

The study domain (Figure 1) is gridded into $96 \times 117$ horizontal grid cells at 4-km resolution with 35 vertical layers. The piecewise parabolic method is used for advection, horizontal diffusion, and eddy vertical diffusion. Chemical reactions are represented using the SAPRC99 mechanism, which includes 72 species and 211 reactions (Carter, 2000). Hourly meteorological inputs for baseline, high-T, and stagnation scenarios are processed from MM5 model outputs (Grell et al., 1994). Time-resolved boundary conditions are derived from the Model for Ozone and Related chemical Tracers (MOZART-4; Emmons et al., 2010). Biogenic emissions are driven by episode-specific meteorological conditions and estimated at hourly resolution using BEIGIS (Scott and Benjamin, 2003). Hourly anthropogenic emissions are provided by the California Air Resources Board for three different years (2000, 2012 and 2022) and are held constant across meteorology. While this approach ensures comparability across scenarios, certain anthropogenic VOC sources, such as evaporative emissions, may also vary with temperature (e.g., Wu et al., 2024), introducing some uncertainty that may affect our results. In our current data, anthropogenic emissions show 45% and 41% reductions for $NO_x$ and VOC during 2000-2012, and 42% and 13% reductions during 2012-2022 (Figures S5-S6).

### 2.4 Adjoint-based Ozone Sensitivity Evaluation

Nine sets of model simulations are performed for three meteorological scenarios (baseline, high-T, stagnation) and three emission scenarios (2000, 2012, 2022). Each scenario combination involves two model runs. First, a CMAQ run is used to evolve pollutant concentration fields by numerically solving a coupled set of partial differential equations of the form

$$\frac{\partial C_i}{\partial t} = -u \cdot \nabla C_i + \frac{1}{\rho} \nabla \cdot (\rho K_i \nabla C_i) + R_i(C) + E_i \qquad (1)$$

where $C_i$ is the concentration of species $i$, $u$ is the vector wind field, $\rho$ is the air density, $K_i$ is the diffusivity tensor, $R_i$ is the net formation rate by chemical reactions, and $E_i$ represents emissions (Byun and Schere, 2006).

Second, a CMAQ adjoint run is performed, which uses data fields saved from the prior CMAQ run and numerically solves another set of partial differential equations in backward time order:

$$-\frac{\partial \lambda_i}{\partial t} = u \cdot \nabla \lambda_i + \nabla \cdot \left( \rho K_i \nabla \frac{\lambda_i}{\rho} \right) + F_i^T \lambda_i + \varphi_i \tag{2}$$

where $\lambda_i$ is called the adjoint variable, $F_i^T$ is the $i$th row of the transposed Jacobian of the chemical reaction rates, and $\varphi_i$ is the forcing term. The mathematical derivation of this adjoint model is detailed elsewhere (e.g., Sandu et al., 2005). Both the forcing term ($\varphi_i$) and the adjoint variable ($\lambda_i$) depend on the choice of receptor. The receptor in this study is defined as the population-weighted 8-hour average $O_x$ ($O_3$ + $NO_2$) concentration over the SJV. Solving the adjoint model yields $\lambda_i(x, y, z, t)$, which represents sensitivity of the receptor to concentration changes of species $i$ at location $(x, y, z)$ and time $t$. Note that these sensitivities are first-order and most reliable for small emission perturbations. The use of three emission scenarios increases confidence that the meteorological impacts reported are qualitatively robust across emissions levels, but quantitative interpretation of these sensitivities as source contribution estimates should be mindful of the underlying first-order accuracy of these results.

We postprocess these $\lambda_i$ outputs to calculate sensitivity of receptor (i.e., SJV population-weighted 8-hour $O_x$) with respect to proportional surface emission changes ($E$) of species $i$ at location $(x, y, z)$ and time $t$:

$$S_i(x, y, t) = E_i(x, y, t) \times \left. \frac{\partial\ SJV\ Ox}{\partial\ E_i} \right|_{(x,y,t)} \tag{3}$$

Notably, these sensitivity values, $\boldsymbol{S_i(x, y, t)}$, are the primary focus of our analysis, representing first-order estimates of ozone contributions from specific source locations, emission periods, and precursor groups (hereafter "*contributions*" and "*sensitivities*" are used interchangeably). Evaluating these sensitivities takes a single adjoint simulation per scenario, whereas a brute-force approach would require over 5,000,000 forward runs to achieve comparable coverage. This highlights the strength of the adjoint method for the purpose of this study.

The sensitivity values are resolved spatially (4 km × 4 km grid cells), temporally (hourly), and by chemical species. We aggregate these values by emission source region (e.g., local, non-local), emission timing (e.g., same-day, prior-day) and precursor group (e.g., $NO_x$, anthropogenic VOC, biogenic VOC) to assess ozone formation regimes and source impacts. We denote a chemical regime as "$NO_x$-limited" when the $O_x$ sensitivity to $NO_x$ exceeds that to anthropogenic VOC, and as "VOC-limited" otherwise. We then evaluate how these sensitivities and chemical regimes shift under varying meteorology, and how such meteorological influences differ in 2000, 2012, and 2022 chemical environments.

Model performance of both CMAQ and CMAQ adjoint has been evaluated in detail for this study domain, as described in Jin et al. (2010) and Wang et al. (2022). The standalone CMAQ model is evaluated by Jin et al. (2010) using ambient observations of ozone and its precursors collected during the Central California Ozone Study (Fujita et al., 2001). Stable model performance is found on both seasonal and episodic timescales, with normalized biases of -5% (1-hour) and 0% (8-hour) for peak ozone values across the SJV. Our configuration options are optimized based on the diagnostic evaluations conducted in Jin et al. (2010). For the CMAQ adjoint model, its sensitivity outputs cannot be directly compared with observations and are instead evaluated against perturbation-based sensitivities. A total of 300 emission perturbations are implemented in CMAQ, each perturbing emissions of one specific species at an individual grid cell. High consistency is seen between adjoint and perturbation-derived sensitivities, with a coefficient of determination at $R^2 = 0.96$ (Wang et al., 2022), confirming the validity of the adjoint framework for quantifying ozone sensitivity to precursor emissions in this domain.

## 3 Results and Discussion

Our research findings are presented in four parts. First, we summarize scenario-induced variations in the SJV ozone pollution level and sensitivity magnitude, with a description of key sensitivity patterns for the baseline meteorology. The second and third sections examine in detail the effects of elevated temperature (high-T) and stagnation, respectively, through comparisons with baseline results. We discuss the underlying drivers and the consistency of meteorological influences over time. Finally, we identify key source locations to target that are effective across meteorological regimes, suggesting that climate-resilient air pollution mitigation strategies are both feasible and essential.

### 3.1 Variation Overview and Baseline Results

Table 2 summarizes SJV $O_x$ levels across all scenario combinations, with $O_x$ spatial distributions shown in Figure S7. Across years, the highest $O_x$ values consistently occur under stagnant conditions (+7-10 ppb relative to baseline), followed by high-T (+4-6 ppb) and then baseline conditions. Long-term emission reductions demonstrate substantial effectiveness, as the $O_x$ receptor decreases by 11-15 ppb from 2000 to 2012, and by an additional 4-6 ppb from 2012 to 2022. Notably, in recent years, meteorology-induced variability has become comparable in magnitude to decadal-scale emission-driven changes. For example, under 2022 stagnant conditions, the SJV $O_x$ burden (62.2 ppb) exceeds that of the 2012 baseline (57.3 ppb), highlighting the role of meteorology in modulating ozone pollution.

**Table 2**. SJV population-weighted 8-hour average $O_x$ concentrations (in ppb) across scenarios.

|  | 2000 | 2012 | 2022 |
|---|---|---|---|
| **Baseline** | 70.6 | 57.3 | 52.2 |
| **High-T** | 76.2 | 61.5 | 55.7 |
| **Stagnation** | 77.3 | 66.5 | 62.2 |

From a sensitivity perspective, the dominant precursor type and source locations vary greatly across scenarios. Figure 2 shows spatiotemporally aggregated sensitivity of $O_x$ to anthropogenic VOC (AVOC) and $NO_x$. Over time, a clear shift in the ozone formation regime is seen, from VOC-limited in 2000 to $NO_x$-limited in 2022, as sensitivity to $NO_x$ continuously rises and sensitivity to AVOC declines. This trend is driven by the more rapid reductions in $NO_x$ compared to AVOC emissions over the past two decades (Figure S5), and aligns with observational evidence from prior studies (Pusede and Cohen, 2012; De Foy et al., 2020).

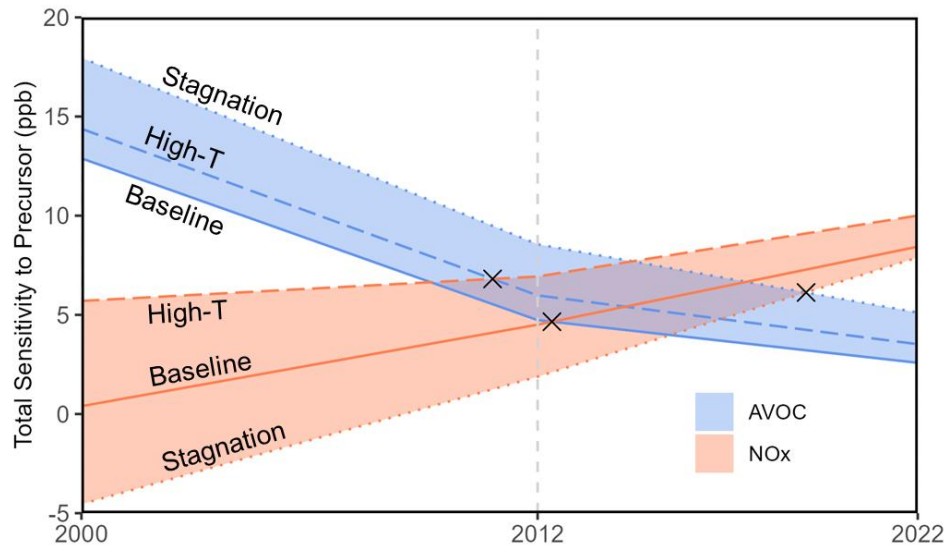

**Figure 2.** Sensitivities of SJV $O_x$ to AVOC (blue) and $NO_x$ (orange) emissions, aggregated over the simulation period and study domain. Color bands show meteorologically-induced variations. Cross marks indicate the points at which the chemical regime switches (when $NO_x$ contributions start to outweigh AVOC contributions).

Although the direction of this shift toward a $NO_x$-limited regime is consistent across meteorological scenarios, the timing of this regime switch (cross marks in Figure 2) can be accelerated or delayed depending on meteorology. Taking the 2012 case as an example, $NO_x$ contribution has dominated under high-T conditions, but AVOC contribution is still 4.5 times $NO_x$ contribution under stagnation. This finding has important implications for other regions, especially those near the transition point, suggesting that episodic meteorological variability can substantially alter the chemical regime and is an essential consideration for designing robust mitigation strategies.

Overall, meteorological impacts on sensitivities are comparable in magnitude to those of decadal-scale emission changes. High-T conditions enhance sensitivity totals by 1.6-5.3 and 0.9-1.5 ppb, respectively, for $NO_x$ and AVOC. Stagnation reduces $NO_x$ sensitivity by 0.5-4.9 ppb and amplifies AVOC sensitivity by 2.5-5.1 ppb. These shifts are in the same order as the sensitivity changes observed from 2000 to 2022 due to emission controls. In the following sections, high-T and stagnation results are compared against baseline to isolate meteorological influences. Unless otherwise specified, patterns reported are consistent across emission scenarios (2000, 2012, 2022) and ranges refer to their spread. Key patterns of baseline sensitivities are summarized here as references for later comparisons.

Under baseline meteorology, anthropogenic contributions are from both local (SJV, 32-49%) and non-local sources (51-68%), with the San Francisco Bay area and Sacramento Valley being the top non-local contributors (Figure S8). Biogenic contributions are relatively small in magnitude, accounting for 15-29% of total sensitivity, primarily from isoprene (81-85%) and monoterpene (11-14%) sources in coastal forests, the Sierra Nevada Mountains, and the northern SJV (Figure S9). Within the SJV, influential source locations – either positive or negative in sign depending on year – are concentrated along the urbanized Highway 99 corridor (see leftmost columns in Figure 4). The prevailing along-valley flow under baseline conditions facilitates pollutant transport towards the southern end of the SJV, amplifying the influence of sources located along plume trajectories that intersect

major cities (e.g., Fresno, Bakersfield; Figure S10). Peak sensitivities occur along the northern and central segments of Highway 99 and in cities nearby, where high emissions coincide with favorable transport pathways toward population centers.

A long-term shift in prevailing chemical regime is evident under baseline conditions, with the most influential source locations (northern and central segments of Highway 99) experiencing a change in sign from negative $NO_x$ contributions in 2000 to positive in 2022. The 2012 scenario marks a transitional case, as the majority (71%) of SJV grid cells show $NO_x$ sensitivities that outweigh AVOC sensitivities, though the opposite remains true in urban and suburban areas (Figure S11). The cities of Fresno and Bakersfield appear to be among the last source areas to undergo the regime switch, as they continue to exhibit negative $NO_x$ sensitivity through 2012.

## 3.2 Temperature Impacts

The comparison between baseline and high-T scenarios features a widespread increase in surface temperatures, with daily average values rising from 24 to 29 °C within the SJV (Table 1). Anthropogenic emissions are held constant for a given year despite the temperature increase, while biogenic VOC (BVOC) emissions are meteorology-driven and increased by ~17%. We also note a reduction in peroxyacetyl nitrate (PAN), suggesting enhanced PAN decomposition and increased $NO_x$ availability, though this effect appears limited given the already warm baseline (Figure S12). Across all years, the high-T scenario leads to increases of 4-6 ppb SJV $O_x$ (Table 2). This is consistent with prior studies that note a positive correlation between temperature and ozone pollution (Bloomer et al., 2009; Pusede et al., 2014; Coates et al., 2016).

From the sensitivity perspective, high-T amplifies SJV $O_x$ sensitivities across years, for both $NO_x$ and VOC, anthropogenic and biogenic, local and non-local sources. Biogenic contributions show a particularly strong relative increase, accounting for 15-29% of total sensitivity under baseline and 22-32% under high-T. For anthropogenic emissions, the spatial partitioning between local and non-local contributions remains similar between baseline and high-T scenarios, with local sources accounting for 31-32%, 40-45% and 47-49% of anthropogenic sensitivities in 2000, 2012, and 2022, respectively. In contrast, high-T effects on the ozone formation regime and the relative importance of AVOC versus $NO_x$ control are more complex, varying with source location and emission scenario as discussed below.

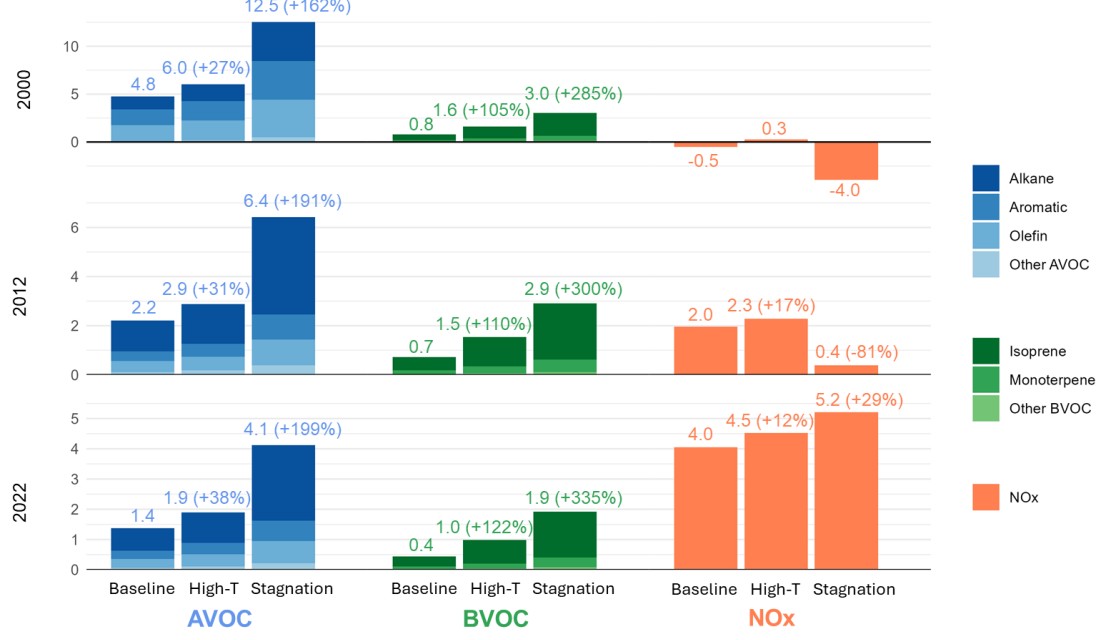

**Figure 3.** Sensitivities (ppb) of SJV $O_x$ to AVOC (blue), BVOC (green) and $NO_x$ (orange) emissions, grouped by species type and scenario. The values shown are summed over the entire SJV air basin. Percentages in parentheses indicate relative changes from baseline. Percent change is not reported for 2000 $NO_x$ because the baseline sensitivity is negative.

Across the SJV, the high-T case increases sensitivity by 0.3-0.8 ppb for $NO_x$, and by 0.5-1.2 ppb for AVOC across emission scenarios (Figure 3). These sensitivity increases are because high-T enhances the generation of $HO_x$ radicals and accelerates temperature-dependent reactions to more efficiently form ozone. In 2000, when the region is characterized by high $NO_x$ emissions and VOC-limited chemistry, elevated $HO_x$ radicals raise the $NO_x$ threshold for regime switch, therefore shifting the chemistry toward $NO_x$-limited conditions. Under the more $NO_x$-limited conditions of 2012 and 2022, high-T conditions

moderately increase the relative importance of AVOC control, as AVOC contributions increase more substantially (+31-38%) than those for $NO_x$ (+12-17%).

Contributions to overall VOC sensitivities by chemical species groupings are shown in Figure 3. AVOC contributions are primarily driven by alkanes, aromatics, and olefins, which together account for >90% of the totals. Under high-T conditions, sensitivities to these three groups increase by similar percentages (+22-40%), with no single group showing disproportionately

larger temperature impacts. On the other hand, BVOC sensitivities are dominated by isoprene and monoterpene, both exhibiting larger relative increases compared to AVOC. Isoprene contributions within the SJV rise by 117-133%, while monoterpene contributions increase by 64-75%. Despite these increases, BVOC contributions remain smaller than AVOC under high-T conditions.

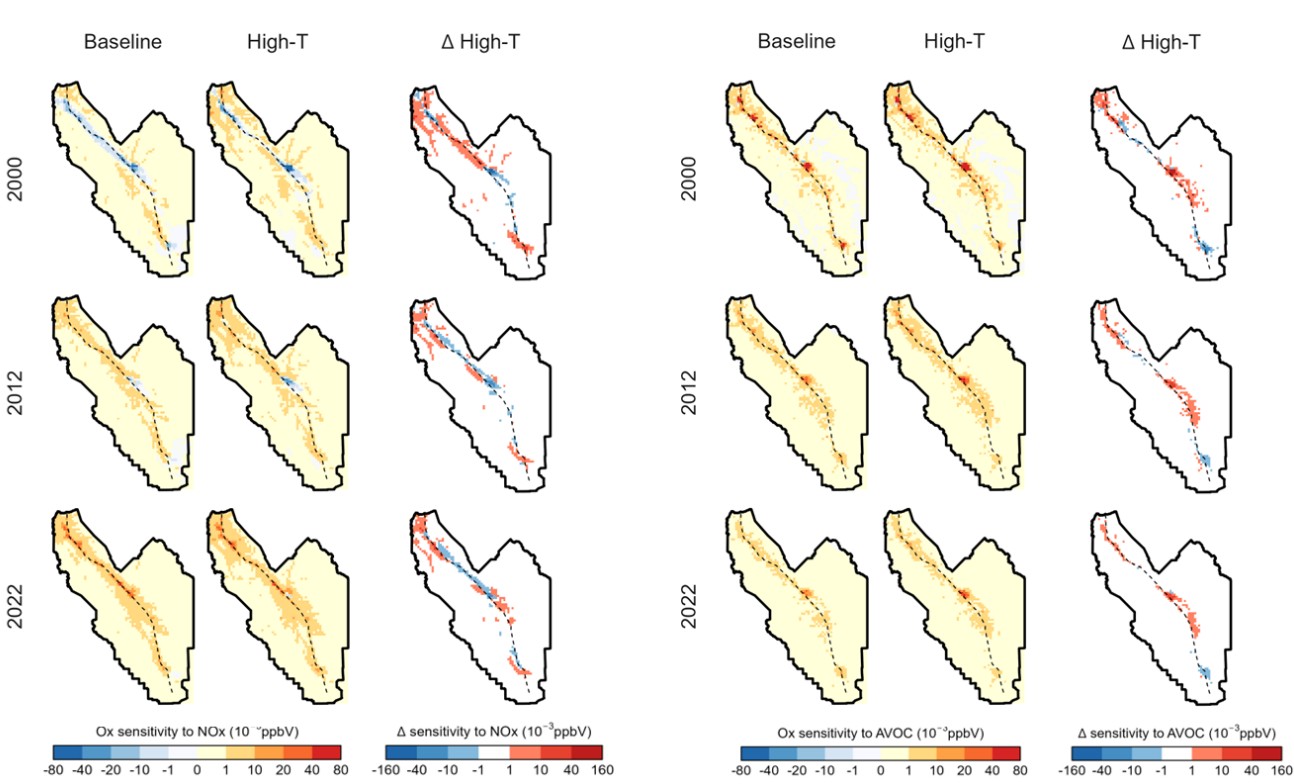

**Figure 4.** Sensitivities of the SJV $O_x$ receptor to (a) $NO_x$ and (b) AVOC emissions under baseline and high-T conditions for 2000, 2012, and 2022. Each panel shows sensitivities under baseline (left column), high-T (middle column), and the difference (high-T minus baseline, right column), with each row corresponding to a different emission inventory year. Higher values indicate larger contributions from the corresponding source locations to the SJV $O_x$ receptor.

Figure 4 compares spatial distributions of $NO_x$ and AVOC sensitivities within the SJV under baseline and high-T conditions. While sensitivity totals for both precursors increase (Figure 3), their enhancements typically occur at different locations (Figure 4). At the grid cell level, temperature-induced changes in $NO_x$ and AVOC sensitivities tend to be opposite in sign, with consistently negative correlations across years (Figure S13). In the southern end of the valley, $NO_x$ sensitivities increase and AVOC sensitivities decline with temperature, while the opposite trend is observed in the middle portion of the SJV valley near Fresno. The northern SJV exhibits a mix of both behaviors.

The optimal source locations for emission control are not substantially shifted by temperature in general, with the only exception being $NO_x$ in the earliest (2000) emission inventory. When ranking SJV grid cells by their individual contributions to the $O_x$ receptor, the correlations between high-T and baseline rankings are R = 0.75-0.97 for $NO_x$, and R = 0.97-0.98 for AVOC (see Figure S14). The lowest correlation coefficient (0.75 for $NO_x$ as of 2000) reflects a shift in chemical regime at certain influential source locations (e.g., Bakersfield and the northern SJV), where high-T drives the chemistry toward a more $NO_x$-limited state and reverses the sign of $NO_x$ contributions, thus elevating these areas from low- to top-priority targets for emission control.

Overall, our findings help bridge the contrasting reports on temperature-driven ozone sensitivity shifts, as reviewed in the introduction. We show that temperature effects depend strongly on baseline chemical and emission conditions. In our domain,

warming promotes a shift towards $NO_x$ limitation when the baseline is VOC-limited (year 2000), consistent with patterns reported in other studies (Nussbaumer and Cohen, 2020; Yang et al., 2021; Huang et al., 2025). In an already $NO_x$-limited regime (year 2022), sensitivitities to both $NO_x$ and VOC rise similarly, so the directional shift is less definitive and may depend on regime definition. Multiple factors (e.g., temperature-dependent VOC emissions, chemical kinetics, water vapor, PAN decomposition) operate simultaneously and may exert opposing influences, with relative importance that varies by location. Thus, temperature effects on ozone sensitivity are expected to vary across different regions, and future studies in other areas are needed. The adjoint-based framework used in this study is broadly applicable, and we recommend its use for evaluating ozone sensitivities to precursor emissions and responses to warming.

### 3.3 Stagnation Impacts

The stagnation scenario is characterized by reduced boundary layer heights, weak surface winds, and diminished coastal inflows. Biogenic emissions are similar to baseline (difference < 5%), and anthropogenic emissions remain constant. Under stagnant conditions, SJV $O_x$ increases by 7-10 ppb relative to baseline (Table 2). Unlike high-T impacts, stagnation markedly alters both the magnitude and spatial distributions of sensitivity. As illustrated in Figure 5, it significantly shifts the relative importance of $NO_x$ versus VOC, anthropogenic versus biogenic, local versus non-local, and prior-day versus same-day sources. While this visualization only shows results for 2022, the patterns discussed below generally hold true across emission scenarios.

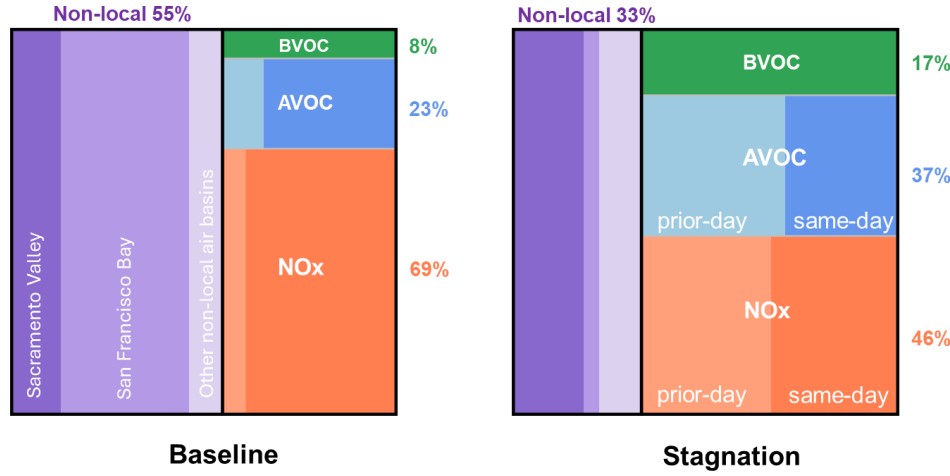

**Figure 5.** Source contributions to SJV $O_x$ under baseline (left) and stagnant (right) conditions for 2022. Each panel shows the relative contributions from non-local sources (purple), as well as local BVOC (green), AVOC (blue) and $NO_x$ (orange) sources within the SJV. Local AVOC and $NO_x$ sensitivities are further attributed to same-day (darker shading) versus prior-day (lighter shading) emissions.

Stagnation significantly amplifies the relative importance of local emission sources. Under baseline conditions, SJV sources (anthropogenic and biogenic combined) account for 29-45% of total sensitivity, but this percentage rises to 56-67% under stagnant conditions. Meanwhile, the influence of San Francisco Bay area sources substantially declines from 32-41% to 4-7%

under stagnation, as there is less inflow to the SJV and therefore reduced Bay area influence on ozone formation (Figure 1d). The other major non-local source area – Sacramento Valley (SV) – shows relatively stable contributions across meteorology, accounting for 13-17% of total sensitivity under baseline and 18-19% under stagnation.

Within the SJV, substantial sensitivity increases are observed for prior-day emissions, due to prolonged residence times and enhanced carryover effects. Under baseline meteorology, $O_x$ sensitivity to local sources is predominantly attributed to same-day emissions (66-82%), followed by emissions from one day prior (15-27%). For stagnant conditions, however, prior-day emissions from one day (37-44%) and two days (17-28%) prior outweigh same-day source contributions to $O_x$ formation (28-36%), highlighting the need for anticipatory emission planning and multi-day control strategies to mitigate stagnation-induced ozone episodes.

Most notably, stagnation consistently drives the ozone chemistry to become more VOC-limited. On a domain-wide level, AVOC sensitivity increases by 2.5-5.1 ppb and $NO_x$ sensitivity decreases by 0.5-4.9 ppb across emission scenarios. Inside the local air basin, stagnation leads to strong enhancements in AVOC and BVOC sensitivities (+160–200% and +285–335%, respectively), whereas $NO_x$ sensitivity either decreases (for 2000 and 2012) or increases modestly (for 2022, when local enhancement effect outweighs the chemical shift). Major AVOC species alkanes, aromatics, and olefins see sensitivity increase by +201-233%, +144-160%, +129-141%, respectively, and major BVOC species isoprene and monoterpene see sensitivity increase by +313-351% and +188-261%. Overall, the relative importance of VOC control substantially rises under stagnation and is comparable to $NO_x$ even in 2022 (Figure 5), which is a clearly $NO_x$-limited case under baseline meteorology. A mechanistic explanation for this regime shift is that elevated $NO_x$ concentrations under stagnation enhance the chain-terminating reaction (OH + $NO_2$ → $HNO_3$) that depletes $HO_x$ radicals (Figure 6). As ozone formation relies increasingly on VOC oxidation to regenerate $HO_x$, sensitivity to AVOC emissions is amplified while the effectiveness of additional $NO_x$ diminishes.

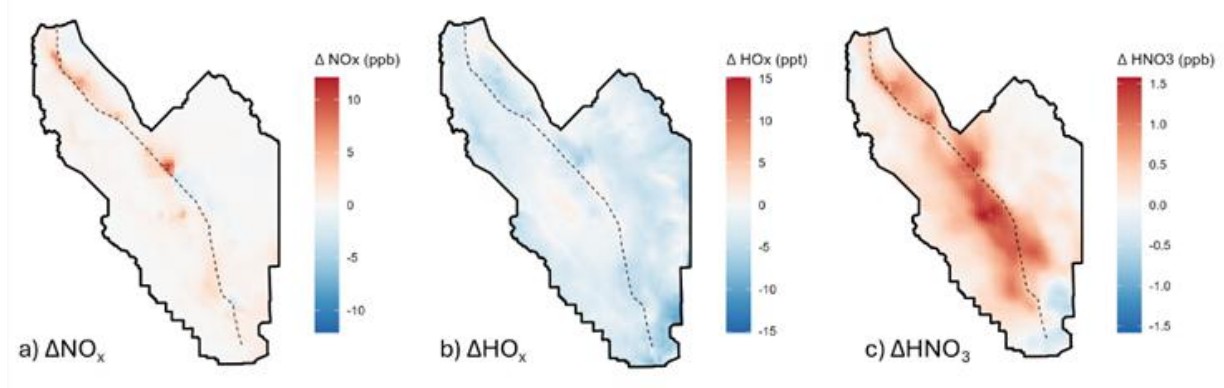

**Figure 6.** Changes in a) $NO_x$, b) $HO_x$ and c) $HNO_3$ concentrations within the SJV under stagnation conditions, relative to baseline, for the 2022 emission scenario.

Figure 7 contrasts spatial sensitivity patterns between baseline and stagnation scenarios. For both precursor types, there is a noticeable expansion of influential source areas under stagnation, especially near the center of the SJV. Under baseline conditions, impactful sources are mainly aligned with along-valley flow trajectories that intersect major cities. During a stagnation event, however, lower boundary layers and weaker winds limit outflows at the southern end, strengthening return flows and eddy recirculation within the valley. A larger spatial range of sources can now interact with urban plumes and

contribute to O$_x$ formation. This transport-driven effect helps explain the modest increase in local NO$_x$ contributions in 2022 (Figure 3), despite an overall shift toward VOC-limited chemistry. Chemistry-driven reductions in NO$_x$ sensitivities dominate in the northern and southern SJV, while transport-driven enhancements are more apparent in the middle portion of the SJV. Together, the net change in SJV NO$_x$ contribution is negative in 2000 and 2012, but becomes slightly positive in 2022 when transport-driven effects outweigh regime shift effects.

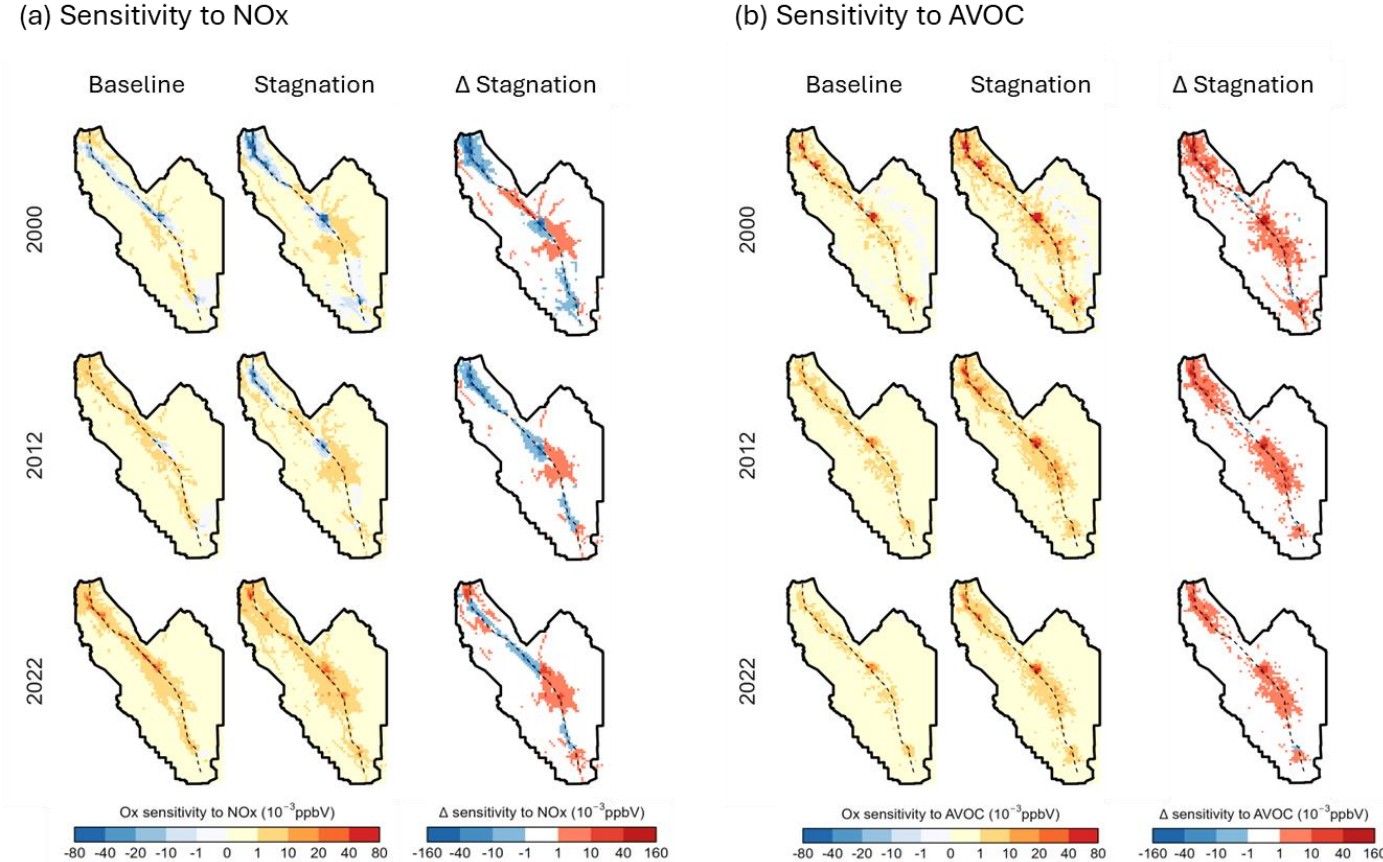

**Figure 7.** Sensitivities of SJV O$_x$ to (a) NO$_x$ and (b) AVOC emissions under baseline and stagnant conditions. Each panel shows sensitivities under baseline (left column), stagnation (middle column), and the difference (stagnation minus baseline, right column), with each row corresponding to different emission inventory scenarios. Higher values indicate larger contributions from the corresponding source locations to SJV O$_x$.

Stagnation can shift optimal source locations, and especially so for NO$_x$ control in VOC-limited (2000) or transitional (2012) environments. The stagnation-induced regime shift toward VOC limitation triggers a sign switch in sensitivity within some major NO$_x$ source regions, and the transport-driven effects lead to emergence of new sensitivity hotspots in the valley center. We find poor correlations between stagnation and baseline rankings of SJV grid cells by their individual NO$_x$ contributions, with R = 0.49, 0.57, and 0.94 in 2000, 2012, and 2022 (Figure S15). These correlation values highlight that stagnation can substantially alter optimal control targets – especially for NO$_x$ – by reshaping both chemical regimes and pollutant transport patterns.

Overall, our findings regarding stagnation impacts align well with existing literature (Biswas and Rao, 2021; Baertsch-Ritter et al., 2004; Jin et al., 2013), while offering a more spatiotemporally detailed evaluation across different emission scenarios. We

find that stagnation consistently shifts ozone chemistry toward VOC-limited conditions, emphasizing the significance of meteorology-driven regime shifts. Since stagnation can co-occur with high temperatures, this may confound the attribution of temperature effects in observational studies. Additionally, stagnation alters the spatial distribution of precursor sensitivities, thereby shifting optimal target locations for emission control.

**3.4 Climate-Resilient Ozone Mitigation**

As demonstrated in prior sections, changes in prevailing meteorological conditions – specifically high temperatures and stagnation – can substantially alter ozone precursor sensitivities, chemical regimes, and the most effective source locations for emission control. These changes highlight that mitigation strategies designed under average conditions may fall short under extreme weather events, which are expected to become more frequent with climate change (Meehl and Tebaldi, 2004; Horton et al., 2014; Hou and Wu, 2016). To support robust control strategy design, Figure 8 identifies SJV grid cells that consistently rank among the top 10% of contributors to $O_x$ formation under all three meteorological scenarios. These maps serve as a screening tool for pinpointing "climate-resilient" source areas where emission reductions consistently yield strong ozone benefits, regardless of weather conditions. Source sector-resolved attribution is also feasible and recommended for future study, but was not undertaken here because our emission dataset lacks sectoral detail.

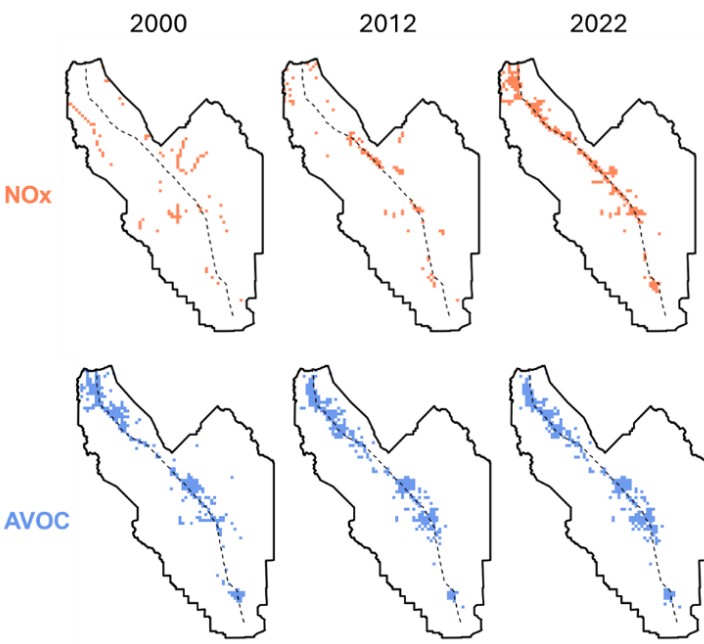

**Figure 8**. Optimal SJV emission source locations to target for control where sensitivities to $NO_x$ (top row) and AVOC (bottom row) emissions consistently rank in the top 10% across meteorological scenarios in 2000, 2012, and 2022. The dashed line represents Highway 99.

Figure 8 reveals spatial differences in the optimal strategies that target $NO_x$ versus AVOC emissions. For $NO_x$, relatively few source locations are impactful across meteorology in early years (2000 and 2012), with only ~2.5% of SJV grid cells consistently ranking among the top 10% contributors. This reflects the high variability of $NO_x$ control effectiveness to meteorological changes. The overlap increases for 2022, with 6.2% of SJV grid cells consistently identified as top 10% contributors, indicating

that the spatial stability of optimal locations for $NO_x$ control improves as the chemical environment transitions toward $NO_x$ limitation. In contrast, the optimal locations for AVOC control are quite stable with respect to meteorological variations. As shown in Figure 8, climate-robust AVOC target locations for emission control account for 7.3-7.6% of grids cells in the SJV across emission scenarios and are consistently distributed along Highway 99.

Over time, we find a growing subset of sources that remain influential across meteorological scenarios, accompanied by increasing spatial convergence in source targets between $NO_x$ and VOC. These trends create opportunities for the SJV to design co-control programs that are both spatially targeted and robust to weather variability. The implications extend beyond the SJV to other regions in VOC-limited or transitional regimes. As anthropogenic emissions decline and ozone chemistry shifts toward $NO_x$ limitation, there is reduced influence of meteorology variability on optimal strategy design. Thus, emission reductions not only bring direct benefits for lowering pollution levels, but also enable the development of climate-resilient control programs to maintain healthy air quality under meteorology extremes.

## 4 Conclusions

This study employs the CMAQ adjoint model to explore how high temperature and stagnant meteorology affect ozone sensitivity to precursor emissions in California's San Joaquin Valley (SJV) under three distinct emission scenarios (2000, 2012, and 2022). While prior work has emphasized the role of meteorology in modulating ozone levels, our results highlight its underappreciated influence on chemical sensitivity, source-receptor relationships, and ultimately the design of optimal ozone mitigation strategies. We find that meteorology-driven sensitivity changes can be comparable in magnitude to those from decadal emission reductions.

High-temperature conditions (+5 °C) broadly enhance ozone sensitivity across precursor types and source regions. Sensitivity to biogenic VOC is more than doubled, despite only ~17% increases in its emissions. In a VOC-limited environment (e.g., the 2000 emission), high-T shifts the ozone regime toward $NO_x$ limitation, altering the effectiveness of $NO_x$ control and changing optimal source locations to target. In contrast, under $NO_x$-limited conditions (e.g., the 2022 emission), optimal source targets remain more stable and the relative importance of VOC control increases.

Stagnation, on the other hand, consistently shifts ozone chemistry toward VOC limitation. VOC contributions can be up to 4 times higher than baseline, while sensitivity to $NO_x$ generally declines or only modestly rises. Impacts of local prior-day sources are substantially amplified under stagnation, as lower boundary layer heights and weaker winds enhance pollutant recirculation and carryover effects. We also find substantial spatial shifts in optimal source targets, though stability improves over time as the region transitions into a $NO_x$-limited regime.

Taking these variations into account, we identify climate-resilient source targets for effective ozone mitigation regardless of meteorology variability. On decadal timescales, we find a wider range of "climate-resilient" targets and a spatial convergence between $NO_x$ and VOC sources over time. As the chemical environment continues to evolve, our findings suggest that incorporating meteorological extremes into mitigation planning becomes both necessary and feasible to sustain long-term ozone air quality goals.

**Competing interests**

Yuan Wang is a member of the editorial board of Atmospheric Chemistry and Physics.

**Acknowledgements**

This research was supported by the California Energy Commission under contract EPC-17-028. Model simulations were conducted on the Lawrencium high performance computing cluster at Lawrence Berkeley National Laboratory. We thank Jeremy Avise at the California Air Resources Board for providing the gridded emission inventory for year 2012, and Wei Zhou, Huy Phan, and Tin Ho for their assistance in emission and meteorological input data processing and model testing.

**Author Contributions**

YhW: conceptualization, methodology, formal analysis, visualization, writing – original draft. LB: methodology, formal analysis, writing – review and editing; YW: formal analysis, visualization, writing – review and editing. LJ: conceptualization, methodology, formal analysis, visualization, writing – review and editing, funding acquisition. RH: conceptualization, methodology, formal analysis, visualization, writing – review and editing, funding acquisition.

**Code and Data Availability**

The original CMAQ adjoint v4.5 model is available at https://people.cs.vt.edu/~asandu/Software/CMAQ_ADJ/CMAQ_ADJ.html. The modified version of the code used in this study can be found in the Supplement of Bastien et al. (2019). Key inputs and output data are provided at https://github.com/yhanw0719/paper_met_o3_adj, and more detailed data are available upon request.

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
