# Peer review of "Temperature and Stagnation Effects on Ozone Sensitivity to NOx and VOC: An Adjoint Modeling Study in Central California"

_EGUsphere, 2025_

## Author Response (AR1)

We thank the reviewers for their constructive and insightful comments, which have helped us improve our manuscript. Detailed responses are provided below in blue, with the reviewers' original comments in *italic black*. Revisions to the manuscript are shown in *italic purple*.
* * *
**Reviewer 1**

**General comments**

*This is a very nice paper telling us the impacts of heatwaves and stagnation on $O_3$-$NO_x$-VOC relationships. It presents some new findings, fits the scope of ACP, and has strong policy implications. I would recommend it to be published after some revisions.*

**Major comments:**

*I think a very important point that could be improved is the separate analysis of heatwave and stagnation. Now we are more and more concerned on "compound events" (e.g., heatwave and stagnation happen simultaneously) and it is believed that such events are likely to increase in a climate-change world. I understand that it may be time-consuming to have new modeling for "compound events", but I recommend that authors to have some analysis and discussions in some ways they prefer.*

We agree that compound extreme events will be increasingly important as climate change progresses. In our study, meteorological regimes were identified via cluster analysis to represent statistically distinct episodes. By design, this approach makes it difficult to isolate compound events, as they typically occur at the boundaries between clusters. While there are prior studies on compound event impacts on ozone pollution, the effect on ozone sensitivity (rather than ozone itself) is a second-order question and remains largely unexplored. As summarized in our introduction, even single-event impacts on precursor sensitivity are not yet so clearly defined. In-depth review/discussion of compound extreme events would require substantial further analysis and is beyond the scope of what we are trying to accomplish in this study.

Our stagnation scenario does exhibit a modest temperature anomaly (~2°C above baseline), reflecting the reality that stagnation tends to limit heat dissipation and results in temperature increases. While this scenario shows some compound characteristics, our sensitivity results indicate that changes are primarily attributable to stagnant flow dynamics rather than temperature increases. We agree that future research that explicitly evaluates compound extremes is relevant and of interest, as the frequency is expected to rise with climate change and they present unique challenges for air quality management.

We have added text in section 2.2 to acknowledge this limitation and recommend it for future study.

*"We acknowledge that compound extremes (e.g., concurrent heat and stagnation) are also of growing importance (Gao et al., 2013; Zhang et al., 2018), but our scenario selection is designed to capture statistically distinct regimes rather than their overlap. Explicit evaluation of compound effects is beyond the scope of this study and is a recommended focus for future work."*

*The modeling finds that stagnation reduces $NO_x$ sensitivity and amplifies AOC sensitivity (e.g., Figure 2). Why are they opposite? Some chemical or meteorological explanations (or both of them) are needed.*

The opposite signs are expected as stagnation shifts ozone formation to a more VOC-sensitive regime. Meteorologically speaking, stagnation leads to NOx accumulation along the major corridor in the northern SJV, increasing the NOx/VOC ratios there (Figure S15). Chemically, this means ozone production becomes less limited by NOx, and more dependent on HOx (i.e., OH + $HO_2$) radicals, which are primarily generated/regenerated from VOC oxidization. Adding NOx into the system both (1) increases the NOx budget, and (2) removes HOx radicals via a chain terminating reaction (OH + $NO_2$ → $HNO_3$). Now in a VOC-limited case, the latter factor becomes more pronounced, thus we see decreased sensitivity to NOx. For the same reason, as ozone formation becomes more dependent on VOC oxidization, its sensitivity to AVOC rises.

We have added text in section 3.3 paragraph 4, as well as Figure 6 to clarify this:

*"A mechanistic explanation for this regime shift is that elevated $NO_x$ concentrations under stagnation enhance the chain-terminating reaction (OH + NO2 → HNO3) that depletes HOx radicals (Figure 6). As ozone formation relies increasingly on VOC oxidation to regenerate HOx, sensitivity to AVOC emissions is amplified while the effectiveness of additional NOx diminishes."*

[Figure]

*Figure 6. Changes in a) NOx, b) HOx and c) HNO3 concentrations within the SJV under stagnation conditions, relative to baseline, for the 2022 emission scenario.*

*I also recommend the authors to add some analysis on PAN since it is an important intermediate for $O_3$ formation, especially by transporting to downwind areas.*

We appreciate the suggestion regarding PAN. Our analysis shows a PAN decrease under high-T conditions as expected, indicating enhanced thermal decomposition and increased local NOx availability. We note that the baseline meteorology is already moderately warm, resulting in relatively low baseline PAN, so the temperature effect via PAN decomposition may not be a dominant driver (ΔPAN at <1ppb level). This PAN discussion has been incorporated into the main text, with text added in section 3.3 and figures added in SI:

*"We also note a reduction in peroxyacetyl nitrate (PAN), suggesting enhanced PAN decomposition and increased NOx availability, though this effect appears limited given the already warm baseline (Figure S12)."*

[Figure]

*Figure S12. Changes in PAN concentration under high-T conditions, relative to baseline, for the 2000, 2012, and 2022 emission scenarios.*

*I am a little bit confused about the opposite sign of NO$_x$ and AVOC sensitivity changes due to temperature increase at grid scale (Fig. 4). Because in Fig.2, it seems that the changes are in the same direction, but in Fig.4, it is completely different. I am not an expert on adjoint approach, but please explain it and have more detailed explanations in the context.*

While both NOx and AVOC sensitivities increase at the regional scale (Figure 2), their changes at the grid cell level (Figure 4) often head in opposite directions. We do not see this difference as a contradiction, but rather it reflects the complexity of temperature impacts. Temperature influences ozone formation through multiple channels (e.g., temperature-dependent emissions, chemical kinetics, PAN decomposition), which can have offsetting effects. The relative importance between these drivers and thus the net impact can exhibit large spatial variability at the grid cell level, due to different source profiles and micro-scale meteorology. For an individual grid cell, the chemical regime may be shifted towards more NOx or VOC limitation, and in either case this leads to opposite signs in sensitivity changes for NOx versus AVOC, as seen in Figure 4. On the other hand, when summed over the entire region (Figure 2), local decreases in NOx or AVOC sensitivity are outweighed by increases elsewhere, resulting in net increases for bother precursor types. This reflects the major response at the regional scale: high-T enhances ozone formation and thus the chemical contribution per unit of precursor emission, therefore the absolute values of NOx and AVOC sensitivity both rise (Figure 2).

*In the introduction part (around line 45-line 70), the authors have a nice literature review on previous studies using different methods with different findings (even opposite). I think the authors should add some discussions to clarify if the findings in this study are different or not and what the key influencing factors are (e.g., research area, models, metrics…). I think a comprehensive comparison would improve the insights of the current study on how to do this kind of research on O$_3$-NO$_x$-VOC relationships in the future.*

*I recommend the authors to have some discussions on how this study would provide new insights for studies in other regions and other scales (since this study focuses on a very specific region), such as East Asia and Europe. Discussing limitations and uncertainties would be appreciated.*

We have added two paragraphs at the end of Section 3.3 (temperature impacts) and Section 3.4 (stagnation impacts) that compare our results with prior literature, discuss key influencing factors, address limitations, and offer recommendations for future research.

For temperature impacts, our findings help bridge the contrasting findings in prior literature, as we find the direction of chemical regime shift depends on baseline

conditions. Since there are multiple influential factors of opposite influences, their relative importance and the overall impacts will strongly depend on regional context. This leads to a key limitation of our study, as we focus on only one area. We note the need for future investigations across a broader range of regions (e.g., Europe and East Asia). It is worth clarifying that the adjoint-based modeling approach we used is generally applicable for any region, any pollution type, and even at larger scales (e.g., GEOS-Chem adjoint for national-scale).

For stagnation impacts, our findings are consistent with prior studies, so the key value of our investigation lies in the diverse emission scenarios used and the level of detail provided. We found that the stagnation-induced shift towards VOC limitation is robust across diverse scenarios. Our analysis also demonstrates how stagnation shifts optimal emission control targets, a finding with direct policy implications that has not been systematically addressed in existing literature. Since the body of literature on stagnation-sensitivity relationships is limited, we strongly recommend increasing attention to this topic, particularly in regions where stagnant meteorology is prevalent or expected to become more frequent due to climate change.

Our in-line revisions are as follows:

*Section 3.2: "Overall, our findings help bridge the contrasting reports on temperature-driven ozone sensitivity shifts, as reviewed in the introduction. We show that temperature effects depend strongly on baseline chemical and emission conditions. In our domain, warming promotes a shift towards NOx limitation when the baseline is VOC-limited (year 2000), consistent with patterns reported in other studies (Nussbaumer and Cohen, 2020; Yang et al., 2021; Huang et al., 2025). In an already NOx-limited regime (year 2022), sensitivities to both NOx and VOC rise similarly, so the directional shift is less definitive and may depend on regime definition. Multiple factors (e.g., temperature-dependent VOC emissions, chemical kinetics, water vapor, PAN decomposition) operate simultaneously and may exert opposing influences, with relative importance that varies by location. Thus, temperature effects on ozone sensitivity are expected to vary across different regions, and future studies in other areas are needed. The adjoint-based framework used in this study is broadly applicable, and we recommend its use for evaluating ozone sensitivities to precursor emissions and responses to warming."*

*Section 3.3: "Overall, our findings regarding stagnation impacts align well with existing literature (Biswas and Rao, 2021; Baertsch-Ritter et al., 2004; Jin et al., 2013), while offering a more spatiotemporally detailed evaluation across different emission scenarios. We find that stagnation consistently shifts ozone chemistry toward VOC-limited conditions, emphasizing the significance of meteorology-driven regime shifts. Since stagnation can co-*

*occur with high temperatures, this may confound the attribution of temperature effects in observational studies. Additionally, stagnation alters the spatial distribution of precursor sensitivities, thereby shifting optimal target locations for emission control."*

**Minor comments:**

*Line 94: Please add references for "The year 2000 represents a more VOC-limited environment, whereas year 2022 reflects cleaner, $NO_x$-limited conditions following major $NO_x$ emission reductions."*

We have added references accordingly:

*"The year 2000 represents a more VOC-limited environment, whereas year 2022 reflects cleaner, NOx-limited conditions following major NOx emission reductions (Pusede and Cohen, 2012; De Foy et al., 2020; Wang et al., 2023)."*

*Line 108: An altitude map of the area would be nice (maybe in SI). It helps readers to understand the wind flows and accumulation of air pollutants.*

We have added an altitude map (Figure S1 in supporting information) and pointed the audience to this map in main text:

*"The valley is bordered by the Sierra Neveda Mountains to the east and Coastal Mountain range to the west (Figure S1)."*

[Figure]

*Figure S1. Surface altitude in the modeling domain.*

*Line 190: The authors write "We denote a chemical regime as "NO$_x$-limited" when the O$_x$ sensitivity to NO$_x$ exceeds that to anthropogenic VOC, and as "VOC-limited" otherwise." So, there is no "transitional regime" defined here? Also, please add a map to tell the readers the spatial distributions of chemical regimes at grid level.*

In our analysis, chemical regimes are classified as either "NOx-limited" or "VOC-limited" at the grid level, based on whether Ox sensitivity to NOx exceeds that to anthropogenic VOC. We did not explicitly define a "transitional regime" in this framework, but we do acknowledge that grid cells with nearly equal sensitivities to NOx and VOC may be considered transitional. We chose not to include a transitional category, as our investigation focuses primarily on the direction and spatiotemporal dynamics of meteorology-induced regime shifts, and introducing an additional regime would add complexity to an already multidimensional analysis. To aid interpretation, the grid-level spatial distributions of NOx- versus AVOC-limited regimes are shown in Figure S11.

[Figure]

Figure S11. Preferred precursor type to target inside the San Joaquin Valley (SJV). A grid cell is colored in blue if its anthropogenic VOC (AVOC) contribution to SJV Ox outweighs its NOx contribution, and colored in orange otherwise. Results shown are for baseline meteorology.

*Figure 3: In 2022, stagnation increases NO$_x$ sensitivity compared to baseline. It is opposite to Figure 2. Why?*

This is because Figure 3 shows sensitivity sums over the SJV air basin (i.e., impacts of local NOx sources). While stagnation generally shifts chemistry toward VOC limitation and reduces overall NOx sensitivity, it simultaneously amplifies the influence of local emissions. In 2022, the local enhancement effect outweighs the chemical effect, resulting in a net increase in local NOx sensitivity as shown in Figure 3. At the domain-wide level,

stagnation consistently decreases NOx sensitivity and increases AVOC sensitivity. We have added clarification in section 3.3 paragraph 4 to avoid confusion:

*"… On a domain-wide level, AVOC sensitivity increases by 2.5-5.1 ppb and $NO_x$ sensitivity decreases by 0.5-4.9 ppb across emission scenarios. Inside the local air basin, stagnation leads to strong enhancements in AVOC and BVOC sensitivities (+160–200% and +285–335%, respectively), whereas $NO_x$ sensitivity either decreases (for 2000 and 2012) or increases modestly (for 2022, when local enhancement effect outweighs the chemical shift)."*

*Line 286: "Under high-T conditions, sensitivities to these three groups increase by similar percentages (+22-40%), with no single group showing disproportionately larger temperature impacts." I think the small differences are because the anthropogenic emissions are unchanged with meteorological conditions, as compared with biogenic emissions. However, there are many studies showing increased anthropogenic emissions as temperatures rise (e.g., Wu et al., 2024). This should be discussed.*

*Wu, W., Fu, T. M., Arnold, S. R., Spracklen, D. V., Zhang, A., Tao, W., … & Yang, X. (2024). Temperature-dependent evaporative anthropogenic VOC emissions significantly exacerbate regional ozone pollution. Environmental Science & Technology, 58(12), 5430-5441.*

We agree that temperature-induced changes in AVOC emissions are not accounted for in the present study. The relevant reference and the following text have been added in section 2.3 to acknowledge this issue:

*"Hourly anthropogenic emissions … are held constant across meteorology. While this approach ensures comparability across scenarios, certain anthropogenic VOC sources, such as evaporative emissions, may also vary with temperature (e.g., Wu et al., 2024), introducing some uncertainty that may affect our results."*
* * *
**Reviewer 2**

*The manuscript "Temperature and Stagnation Effects on Ozone Sensitivity to NOx and VOC: An Adjoint Modeling Study in Central California" quantifies the sensitivity of ozone exposure to NOx and VOC emissions in the San Juan Valley over different emission scenarios and temperature/meteorological conditions. The authors determine how these sensitivities*

*change with respect to temperature due to a changing climate while also accounting for emission reductions. While the analysis presented provides a strong narrative with important results, the paper would benefit from additional work on comparing the choice of methods and evaluation of the results. After the detailed comments (which constitute minor revisions) below are addressed, I would recommend the article for publication.*

**Main Comments**

*Line 84: While not exactly the same as the methods used here, there have been prior studies of the role of meteorological factors on O3 sensitivities using adjoint modeling. The CMAQ adjoint paper (Hakami et al., 2007) included explicit calculation of sensitivities with respect to temperature (via the role of temperature on chemical kinetics); this idea was expanded upon in Zhao et al. (GRL, 2013) which examined the estimation of ozone climate penalties using adjoint sensitivities. Works such as Park et al. (Atmos. Environ., 2018) investigated both chemical and meteorological influences on ozone using adjoint sensitivities. Lastly, while not a study of meteorological impacts, a paper that does though seem pertinent to the use of adjoint modeling to explore variability in O3 isopleths is that of Ashok and Barrett (Atmos. Environ., 2016). In summary, there may be more prior work on this sort of topic using adjoint sensitivity analysis than the authors have indicated, though I still recognize that their particular approach does seem to be unique.*

We have incorporated discussion of Hakami et al. (2007), Zhao et al. (2013), Park et al. (2018), and Ashok and Barrett (2016) to reflect prior use of adjoint modeling to the study of ozone–meteorology interactions.

Our work goes further by addressing a second-order question: how meteorological variability modifies ozone sensitivity to precursor emissions, not just effects on ozone levels themselves. Prior adjoint studies have made important first-order contributions, for example, by examining how ozone exposure metrics respond to meteorological parameters such as temperature. While some of these studies include discussions that hint at second-order effects, prior work has not as far as we can tell assessed how ozone-to-precursor sensitivities themselves change across meteorological regimes. This second-order framing is central to our study. We aim to inform how optimal emission control strategies may need to be adapted under varying meteorological conditions. A paragraph has been added in the introduction to acknowledge the reviewer's suggested studies while clarifying how our work complements and extends this existing body of research:

*"To our knowledge, no prior studies have systematically investigated meteorological impacts on adjoint-based $O_3$-$NO_x$-VOC sensitivities. This is a second-order sensitivity analysis: how ozone sensitivity, rather than ozone itself, responds to meteorological*

*changes. Existing adjoint literature has laid important foundations at the first-order level. Hakami et al. (2007) evaluated ozone exposure sensitivity to local temperatures through chemical kinetics. Zhao et al. (2013) advanced this work to estimate ozone climate penalties in Canada and the U.S. using adjoint-derived temperature sensitivities, considering both direct chemical kinetic impacts and indirect effects through biogenic emissions and water vapor. They found high-$NO_x$ polluted regions were particularly responsive to temperature. Park et al. (2018) assessed $O_3$-$NO_x$-VOC sensitivities in Daegu, Korea, attributing 62% of total contributions to meteorology-driven $O_3$ transport and 38% to chemical reactions. Ashok and Barrett (2016) constructed ozone exposure isopleths in the U.S. using adjoint sensitivities, with discussion of seasonal variability bearing relevance to second-order impacts. While these studies highlight the potential for meteorology to affect ozone regimes and emission control strategies, our work seeks to directly and systematically isolate this effect across representative meteorological regimes."*

*Section 2.4: Please state the temporal resolution used for the sensitivities and how they relate to emissions – are sensitivities calculated hourly, before being aggregated temporally? Are all emissions also hourly, or are some constant throughout the day (which could pose an issue given the analysis here)?*

Yes, both our emissions and adjoint sensitivities are resolved hourly, so diurnal variability is captured. We report sensitivity values after temporal aggregation, so they represent sensitivity to sustained emission changes over the aggregation window. To clarify this in the manuscript, we have added sentences in Sections 2.3 and 2.4:

*"Biogenic emissions are driven by episode-specific meteorological conditions and estimated at hourly resolution using BEIGIS (Scott and Benjamin, 2003). Hourly anthropogenic emissions are provided by the California Air Resources Board for three different years (2000, 2012 and 2022) and are held constant across meteorology."*

*"The sensitivity values are resolved spatially (4 km × 4 km grid cells), temporally (hourly), and by chemical species. We aggregate these values by emission source region (e.g., local, non-local), emission timing (e.g., same-day, prior-day) and precursor group (e.g., NOx, anthropogenic VOC, biogenic VOC) to assess ozone formation regimes and source impacts."*

*Section 2.4, Paragraph 3: It's important to state the limitations of assuming a first-order contribution of emissions to Ox, especially given the nonlinearity in the underlying*

*chemical mechanisms. While less pertinent in the year-to-year comparative analyses, results discussing peak sensitivities and spatial contributions do need this qualification.*

We agree that first-order sensitivities have limitations and must be interpreted where linearity holds. As the reviewer notes, our year-to-year comparisons increase confidence that meteorological impacts on sensitivities are robust with respect to emission variability. However, quantitative use of individual sensitivity values as source-contribution estimates is subject to first-order limitations, and we now caution readers about this issue. We have clarified this in Section 2.4 with the following text:
*"Note that these sensitivities are first-order and most reliable for small emission perturbations. The use of three emission scenarios increases confidence that the meteorological impacts reported are qualitatively robust across emissions levels, but quantitative interpretation of these sensitivities as source contribution estimates should be mindful of the underlying first-order accuracy of these results."*

*Section 2.5, Paragraph 5: Please provide some statistics from the studies you mentioned that analyzed CMAQ performance, like mean bias and standard deviation of ozone concentrations over, if possible, the SJV, just to give context on how well the model calculates ozone exposure.*

Sure, we have revised following text to provide evaluation statistics on CMAQ performance in section 2.5:

*"The standalone CMAQ model is evaluated by Jin et al. (2010) using ambient observations of ozone and its precursors collected during the Central California Ozone Study (Fujita et al., 2001). Stable model performance is found on both seasonal and episodic timescales, with normalized biases of -5% (1-hour) and 0% (8-hour) for peak ozone values across the SJV."*

*Section 3.3, paragraph 3: Could this increase in VOC sensitivity be due to less transport of VOCs from outside of SJV in Stagnant conditions that could otherwise lead to ozone formation? Is that something you could quantify?*

We interpret this comment as suggesting that the rise in local VOC sensitivity under stagnation may stem from local VOCs comprising a larger share of the ozone-relevant precursor pool. This is indeed a contributing factor: the "effective" fraction of VOCs that are locally sourced does rise (evident from the rise in their shares of total VOC sensitivity from ~30-40% under baseline conditions to ~60-70% under stagnation). However, two other

factors are also worth noting. Firstly, the domain sum of VOC sensitivity increases (section 3.3, paragraph 4), indicating not just a redistribution from non-local to local sources but an overall shift toward a more VOC-sensitive chemical regime. Secondly, stagnation not only reduces inflows but also outflows at the southern end of the SJV, increasing pollutant recirculation and residence times. This likely increases the photochemical utilization of local emissions that would otherwise be exported before reacting. While we cannot definitively quantify the contribution of each factor, we believe all three – enhanced local VOC fraction (paragraph 2), longer residence time (paragraph 3), and regime shift (paragraph 4) – jointly explain the observed increase in local VOC sensitivity under stagnation.

*Sentence spanning lines 370-371: Cite the fact that extreme weather events will increase in frequency with climate change.*

We have added citations in section 3.4 paragraph 1 to support this statement:

*"… mitigation strategies designed under average conditions may fall short under extreme weather events, which are expected to become more frequent with climate change (Meehl and Tebaldi, 2004; Horton et al., 2014; Hou and Wu, 2016)."*

*Section 3.4: The spatial results could be supported with details on if NOx and AVOC sectors align for most optimal controls, like if road transport reductions dominate in rural regions versus industrial emissions in the cities, or it is just one sector that dominates. If CARB's inventory provides sectoral emission distributions I would consider adding this analysis.*

We agree that sector-resolved analysis would strengthen the spatial results, but unfortunately, the gridded inventory available to us is aggregated across sectors (i.e., lacks source category tags) and we were unable to obtain a sector-resolved version from the original provider. We acknowledge this as a limitation we would very much like to address, but are unable to do so in the current manuscript. Following text has been added in section 3.4 paragraph 1:

*"Source sector-resolved attribution is also feasible and recommended for future study, but was not undertaken here because our emission dataset lacks sectoral detail."*

*Section 3: Overall, it is unclear what the range in result values pertains to. Are these confidence intervals? Or the range in values in the SJV?*

The ranges reported in Section 3 refer to the spread across the three emission scenarios (2000, 2012, 2022), not confidence intervals. We have added clarifications in Section 2.1 to make this explicit:

*"(In the following sections, high-T and stagnation results are compared against baseline to isolate meteorological influences). Unless otherwise specified, patterns reported are consistent across emission scenarios (2000, 2012, 2022) and ranges refer to their spread."*

*It would be beneficial to quantify the advantage of using the adjoint model as opposed to a "brute force" method either in the discussion or where the adjoint method is introduced, something like "where the brute force method would require X simulation runs, our adjoint model only requires 9 to provide the same results"*

We have added text in section 2.4 paragraph 3 to highlight this strength of the adjoint method:

*"Evaluating these sensitivities takes a single adjoint simulation per scenario, whereas a brute-force approach would require over 5,000,000 forward runs to achieve comparable coverage. This highlights the strength of the adjoint method for the purposes of this study."*

**Minor Edits**

*Figure 3: missing percentages on 2000 NOx, is this on purpose?*

Yes, the omission is intentional. Percent changes are computed relative to the baseline value, and the NOx baseline sensitivity is negative in 2000, making percent change values misleading. As the absolute sensitivities are already reported, we chose not to show percent changes for this case.

We have added a clarifying note to the Figure 3 caption explaining this choice:

*"(Percentages in parentheses indicate relative changes from baseline). Percent change is not reported for 2000 $NO_x$ because the baseline sensitivity is negative."*

*Figure 5: Consider including geographic regions in "non-local" category, could be interesting to see how, e.g., SF Bay's contribution changes in a graphical presentation with respect to other areas, or to see if there is an outside region that contributes most in Baseline but becomes negligible in Stagnation.*

We have updated Figure 5 to show geographical breakdown in the "non-local" category. The most notable pattern is the sharp reduction in SFB contributions under stagnation, as discussed in text following Figure 5. For the "other non-local air basins" (e.g., Mountain Counties), their individual impacts (and their changes under stagnation) are much smaller in magnitude compared to Sacramento Valley and SF Bay area, so they are grouped in the "Other" category and not discussed separately.

[Figure]

*"... the influence of San Francisco Bay area sources substantially declines from 32-41% to 4-7% under stagnation, as there is less inflow to the SJV and therefore reduced Bay area influence on ozone formation."*

*Line 29: Use of Oxford comma here, but omitted on Line 24, please keep consistent. Can also omit "by" at the beginning of each dependent phrase.*

We have now applied the Oxford comma consistently on line 24 and removed "by" at the beginning of the dependent phrases on line 28.

*Line 35: Repetitive "while" use, I'd try to use more unique wording, or omit line all together since it is not necessary. Introduction, Paragraph 2: Seems out of place. I would move to after Paragraph 3 to introduce ozone formation mechanisms and terminology before its sensitivities to model parameters.*

We have moved the former Paragraph 2 to follow Paragraph 3 and revised the text to avoid repetitive "while" use and improve the logic flow. The revised text now reads:

*"Although meteorological controls on ozone concentrations are well studied, far less work has examined how meteorology modulates $O_3$-$NO_x$-VOC sensitivities and thus the effectiveness of emission control programs."*

*Line 40: HOx is never defined, either define it or simply say "hydrogen oxide radicals" since HOx is not used that often in the manuscript.*

Thanks for catching this. We have now revised it to "*hydrogen oxide ($HO_x$) radicals*".

*Section 2.3/2.4: The choice to use pop-weighted 8-hr average Ox as opposed to just O3 should be stated here.*

We added following text at the end of the first paragraph in section 2.3 to justify this receptor choice:
*"We use $O_x$ rather than $O_3$ because $O_x$ is conserved under rapid NO-$O_3$ titration, providing a more stable measure of the photochemical oxidant burden for evaluating its sensitivity to precursors."*

*Line 190: Forgot to make subscript x on Ox.*

Corrected.